# Plant Growth-Promoting Bacteria as an Emerging Tool to Manage Bacterial Rice Pathogens

**DOI:** 10.3390/microorganisms9040682

**Published:** 2021-03-26

**Authors:** Mohamad Syazwan Ngalimat, Erneeza Mohd Hata, Dzarifah Zulperi, Siti Izera Ismail, Mohd Razi Ismail, Nur Ain Izzati Mohd Zainudin, Noor Baity Saidi, Mohd Termizi Yusof

**Affiliations:** 1Department of Microbiology, Faculty of Biotechnology and Biomolecular Sciences, Universiti Putra Malaysia, Serdang 43400, Selangor, Malaysia; syazwanngalimat@gmail.com; 2Department of Plant Protection, Faculty of Agriculture, Universiti Putra Malaysia, Serdang 43400, Selangor, Malaysia; erneeza@upm.edu.my (E.M.H.); dzarifah@upm.edu.my (D.Z.); izera@upm.edu.my (S.I.I.); 3Institute of Tropical Agriculture and Food Security, Universiti Putra Malaysia, Serdang 43400, Selangor, Malaysia; razi@upm.edu.my; 4Department of Crop Science, Faculty of Agriculture, Universiti Putra Malaysia, Serdang 43400, Selangor, Malaysia; 5Department of Biology, Faculty of Science, Universiti Putra Malaysia, Serdang 43400, Selangor, Malaysia; ainizzati@upm.edu.my; 6Department of Cell and Molecular Biology, Faculty of Biotechnology and Biomolecular Sciences, Universiti Putra Malaysia, Serdang 43400, Selangor, Malaysia; norbaity@upm.edu.my

**Keywords:** *Oryza sativa*, bacterial rice pathogens, plant growth-promoting bacteria, biocontrol agents, bioformulations

## Abstract

As a major food crop, rice (*Oryza sativa*) is produced and consumed by nearly 90% of the population in Asia with less than 9% produced outside Asia. Hence, reports on large scale grain losses were alarming and resulted in a heightened awareness on the importance of rice plants’ health and increased interest against phytopathogens in rice. To serve this interest, this review will provide a summary on bacterial rice pathogens, which can potentially be controlled by plant growth-promoting bacteria (PGPB). Additionally, this review highlights PGPB-mediated functional traits, including biocontrol of bacterial rice pathogens and enhancement of rice plant’s growth. Currently, a plethora of recent studies address the use of PGPB to combat bacterial rice pathogens in an attempt to replace existing methods of chemical fertilizers and pesticides that often lead to environmental pollutions. As a tool to combat bacterial rice pathogens, PGPB presented itself as a promising alternative in improving rice plants’ health and simultaneously controlling bacterial rice pathogens in vitro and in the field/greenhouse studies. PGPB, such as *Bacillus*, *Pseudomonas*, *Enterobacter*, *Streptomyces*, are now very well-known. Applications of PGPB as bioformulations are found to be effective in improving rice productivity and provide an eco-friendly alternative to agroecosystems.

## 1. Introduction

Rice (*Oryza sativa*) is a staple food in Asia and parts of African countries. Over 90% of the world’s rice is produced in the Asian region and consumed as the main food source. With an ever-increasing human population, it is challenging to ensure food security for the general population. To counter the demand, rice production needs to be increased to fulfill this need [1]. However, throughout the growing season, reduction in rice yields has been detected and are caused by a variety of phytopathogens including bacteria. Rice diseases caused by bacteria are the main constraint towards sustainable productivity of rice [2]. Up until now, extensive work has been carried out on the management of bacterial diseases of rice caused by bacterial species belonging to the genus *Xanthomonas* [3,4,5]. With an awareness to reduce chemical usage to decrease the proliferation of pathogenic microbes, biocontrol is a promising strategy to combat phytopathogens [6].

In plant rhizosphere, plant growth-promoting bacteria (PGPB) are indigenous. PGPB display beneficial effects on their host plant and play a major role in the biocontrol of phytopathogens [7]. With rice, diverse types of bacterial genera, including *Pseudomonas*, *Bacillus*, *Enterobacter*, *Alcaligenes*, *Arthrobacter*, *Azospirillum*, *Azotobacter*, *Burkholderia*, *Klebsiella*, *Rhizobium*, and *Serratia*, have all been potentially considered as PGPB as characterized in vitro [8]. Characterization of PGPB is based on their ability to stimulate plant growth, which involves multiple mechanisms, including direct and indirect mechanisms [9]. Direct mechanisms involve nitrogen fixation, mineral (e.g., phosphorus and iron) solubilization, siderophore production, and phytohormone production (e.g., auxins, cytokinins, gibberellins, and ethylene). Meanwhile, indirect mechanisms are mainly due to the biocontrol activities of PGPB in responding to the biotic stress by producing antibiotics. In addition, PGPB also have a role in the management of abiotic stresses, such as salinity and drought [10]. Thus, to enhance rice productivity, as well as for biocontrol against phytopathogens, the usage of PGPB is needed.

Applications of PGPB on rice have shown notable successes. A plethora of PGPB from the genera *Bacillus* [11], *Pseudomonas* [12], *Enterobacter* [13], and *Streptomyces* [14] were reported to give positive effects on rice plant’s health and growth. Bacterial species, such as *Pseudomonas fluorescens* [15,16], and a bacterial consortium [17] have shown their applicability to enhance rice yields in a form of bioformulations. Studies indicated that PGPB can act as bioinoculants as they promote plant growth, health, and yield [18,19,20]. Currently, PGPB also have been found acting against bacterial rice pathogens [21,22,23]. Hence, PGPB inoculation is emerging as an effective method to combat bacterial rice pathogens for enhancing rice production through eco-friendly approaches. In this review, an overview of bacterial rice pathogens is described. We have discussed PGPB-mediated functional traits including biocontrol of bacterial rice pathogens and enhancement of rice plant’s growth. Future potential uses of PGPB in enhancing rice productivity in the form of bioformulations are also projected.

## 2. Bacterial Rice Pathogens

Rice diseases caused by bacteria are a major bottleneck towards sustainable productivity of rice, especially in Asia and parts of the African countries. In severe epidemics, reduction in rice has reached more than 60% and millions of hectares of rice are infected annually [24,25,26]. Bacterial pathogens, such as *Xanthomonas oryzae* pv. *oryzae*, *Xanthomonas oryzae* pv. *oryzicola*, *Burkholderia glumae*, and *Burkholderia gladioli*, are spread rapidly and sporadically under favorable conditions and cause tremendous obstacles to rice production [27,28]. Bacterial pathogens are easily transmitted from infected plants that travel through the water and spread to the roots and leaves of neighboring plants. The spreading of bacterial pathogens can also be transmitted from contaminated or infected seeds to the emerging seedlings [29].

Bacterial pathogens infect the rice plant at all parts including the seed, foliar, leaf sheath, grain, culm, and root (Table 1). Numerous bacterial pathogens that are reported to cause diseases in rice belong to the genera *Xanthomonas*, *Burkholderia*, *Pseudomonas*, *Pantoea*, *Erwinia*, *Acidovorax*, *Dickeya*, and *Enterobacter*. Bacterial species belonging to the genus *Xanthomonas*, including *X. oryzae* pv. *oryzae* (the causal agent of Bacteria Leaf Blight) and *X. oryzae* pv. *oryzicola* (the causal agent of Bacterial Leaf Streak), are well-known bacterial diseases of rice. It is noteworthy that studies in recent years have indicated numerous bacterial species from the genus *Burkholderia* and *Pantoea* as the next major pathogens of rice [4,29,30]. 

With regard to bacterial pathogenicity, plant pathology and genomic studies have revealed the detection of virulence factors including degradative enzymes, extracellular polysaccharides, and components of quorum sensing signaling molecules, which are in-volved in the communication between the host and pathogen that contribute to rice dis-eases [31,32,33]. This conveys a sense of continuing excitement in the field of molecular plant pathology to control bacterial rice pathogens. To date, the main approaches to control bacterial rice pathogens include the production of disease-resistant rice varieties [34,35,36]; modification in cultural practices [37,38]; use of natural products or botanical extracts [39,40,41]; use of conventional and non-conventional chemicals [42,43]; coevolution analysis of the pathogen virulence and the host resistance genes [44,45]; transcriptomic analysis of pathogen along rice development [46,47]; improvement of diagnostic tools in the field for early detection of infectious diseases [48]. 

However, the effectiveness of these approaches is somehow inefficient due to the polymorphisms and chemical resistance developed in virulent strains [49]. Interestingly, biocontrol strategies by implementing PGPB could be a possible alternative in controlling rice pathogens, which involves the application of disease-suppressive bacteria to control pathogens and improve plant health.

**Table 1 microorganisms-09-00682-t001:** Bacterial rice pathogens and related diseases.

Diseases		Bacterial Pathogens	References
Seedling	Seedling blight	*Burkholderia plantarii*	[50]
	Bacterial Brown Stripe of Rice (BBSR)	*Pseudomonas syringae* pv. *panici*	[51]
		*Acidovorax avenae* subsp. *avenae*	[52]
Foliar	Bacterial Blight (BB) or Bacteria Leaf Blight (BLB)	*Xanthomonas oryzae* pv. *oryzae*	[53]
		*Pantoea ananatis*	[54]
		*Pantoea stewartii* subsp. *indologenes*	[55]
		*Pantoea stewartii*	[54]
		*Pantoea agglomerans*	[56]
	Bacterial Leaf Streak (BLS)	*Xanthomonas oryzae* pv. *oryzicola*	[57]
	Halo blight	*Pseudomonas syringae* pv. *oryzae*	[58]
Leaf sheath and grain rot	Sheath brown rot	*Pseudomonas fuscovaginae*	[59]
	Sheath rot	*Pseudomonas syringae* pv. *syringae*	[60]
	Bacterial Panicle Blight (BPB)	*Burkholderia glumae* or *Burkholderia gladioli*	[29]
	Bacterial palea browning	*Erwinia herbicola*	[61]
		*Pantoea ananatis*	[62]
		*Enterobacter cloacae*	[63]
Culm and root	Bacterial foot rot	*Erwinia chrysanthemi*	[64]
		*Dickeya zeae*	[65]

## 3. An Overview of In Vitro Characterizations of Promising PGPB

Rhizosphere is the soil surrounding plant roots that are rich in nutrients and a potent habitat for microbes that thrive on root exudates known as rhizodeposits [66]. Rhizodeposits comprise various compounds that aid the lubrication and nutrient acquisition of plants [67]. Rhizodeposits act as chemo attractants that welcome large and diverse microbial communities living in the rhizosphere to multiply the roots or adjacent rhizospheric soil [68]. PGPB are a specific category of microbes that are beneficial to the plant or are involved in some positive plant–microbe interactions [69]. The mechanisms of plant–microbe interactions for a successful PGPB to enhance plant growth have been reviewed previously [70,71,72]. As suggested by Kloepper, successful PGPB are characterized by three inherent distinctiveness: (i) must be proficient to colonize the root surface; (ii) must survive, multiply, and compete with other microbiota, at least for the time needed to express their plant growth-promoting activities; (iii) must promote plant growth [73]. Other than being able to colonize plant roots and promote plant growth, PGPB also simultaneously act as biocontrol agents, biofertilizers, phytostimulators, rhizoremediators, and biopesticides [74]. It is noteworthy to highlight that the usage of PGPB as biological agents offered various promising advantages, including enhancement in crop yield and a decrease in disease occurrence [75].

PGPB’s ability as a biocontrol agent is linked to how they improve plant growth and suppress phytopathogens either by direct and or by indirect mechanisms (Figure 1). The characterization of plant growth-promoting mechanisms of PGPB can be determined in vitro (Table 2). Direct mechanisms include the balance of plant growth regulators. PGPB release plant growth regulators that are integrated into the plant and act as a sink of plant-released hormones. Subsequently, this induces the plant’s metabolism, thus leading to an improvement in the plant’s adaptive capacity. In this mechanism, PGPB facilitate resource acquisition such as nitrogen, phosphorus, and essential minerals through biological nitrogen fixation, phosphate solubilization, and iron sequestration by siderophores. PGPB also modulate phytohormone levels such as indole-3-acetic acid (IAA), cytokinins, and gibberellins to promote plant growth [76]. IAA is an auxin produced by PGPB that plays a role in stimulating both rapid (e.g., cell elongation) and long-term (e.g., cell division and differentiation) responses in plants [77]. Similar to IAA, cytokinins influence plant physiological and developmental processes. Plant responses to exogenous applications of cytokinins resulted in enhanced cell division and root development and formation [78]. Gibberellins are important phytohormones that influence the developmental processes in higher plants including, seed germination, stem elongation, flowering, and fruit setting [79]. 

Indirect mechanisms require the involvement of the plants’ defensive metabolic processes that respond to the signal sent from the PGPB. The mechanisms include: (i) induced systemic resistance (ISR) to plant phytopathogens (biotic stress), and (ii) protection against environmental stress (abiotic stress) [97]. In this mechanism, PGPB mediate the production of antimicrobial metabolites under biotic stress by responding to the rhizospheric competition for nutrients and niche exclusion. PGPB produced antimicrobial metabolites such as hydrogen cyanide (HCN), cyclic lipopeptides (CLP), 2,4-diacetylphloroglucinol (DAPG), pyrrolnitrin, pyoluteorin, and phenazines, which are used to inhibit the growth of competing microbes [98]. Interaction of PGPB with plant roots enhances plant resistance against some microbes including pathogenic bacteria, fungi, and viruses. This phenomenon is termed as induced systemic resistance (ISR). Many individual bacterial components, such as siderophores, CLP, DAPG, and volatile organic compounds (VOCs), including acetoin and 2,3-butanediol, act as an elicitor of ISR [99]. Moreover, ISR involves ethylene (a phytohormone that governs plant growth and development) signaling that stimulates the host plant’s defense responses against a variety of phytopathogens [100]. Interestingly, PGPB that produce 1-aminocyclopropane-1-carboxylate (ACC) deaminase enzyme are able to facilitate plant growth and development by lowering ethylene levels through degradation of ACC to ammonia and α-ketobutyrate. Therefore, the PGPB containing ACC deaminase have the potential to reduce abiotic stress by decreasing ethylene levels [101].

## 4. The PGPB as Biocontrol Agent

Biocontrol is a promising strategy to control phytopathogens, which could be an alternative for chemical fertilizers and pesticides. The implementation of PGPB as a biocontrol agent to inhibit the growth of phytopathogens has become widespread due to environmental concerns. This strategy has received great attention as it provides a safe, inexpensive, long-lasting, and environmentally friendly alternative [102]. Bacteria from the genera *Bacillus*, *Pseudomonas*, *Enterobacter*, and *Streptomyces* were all extensively studied as biocontrol agents as soon as their antagonistic activity against rice pathogens was recognized. Apart from these four groups, other genera are highlighted in the section below.

### 4.1. Bacillus spp.

*Bacillus* spp. are Gram-positive bacteria belonging to the phyla Firmicutes, class Bacilli, and family Bacillaceae. It can be characterized as rod-shaped and endospore-forming bacteria. The ability to produce endospores when facing harsh conditions enabled this bacterial species to survive in various habitats including animal feces [103], bee products [104], soil [105], food [106], and aquatic environments [107]. *Bacillus* spp. as PGPB have been proven to confer numerous advantages in the agricultural sector [108]. There are three main contributions of *Bacillus* spp. in rice: (i) increase yield; (ii) improve tolerance to abiotic stresses; (iii) decrease in disease occurrence. 

The colonization of *Bacillus* spp. on crop roots caused an increase in crop yields [75]. Evidently, rice root associated with *Bacillus* was found to improve growth, yield, and zinc (Zn) translocation of Basmati rice. Basmati-385 and Super Basmati rice yields were improved by more than 22% and 18%, respectively, upon inoculation with Zn-solubilizing strains that were identified as *Bacillus* spp. by 16S rRNA gene analysis [109]. Growth and yield were also found to be improved upon a new Egyptian rice line, GZ9461-4-2-3-1, inoculated with a consortium of PGPB containing *Bacillus subtilis, Pseudomonas fluorescens*, and *Azospirillum brasilens*. The integration of inorganic fertilizers with a consortium of PGPB positively affected rice yields and contributed to reducing chemical nitrogen fertilizers by 25% [110]. Moreover, nursery application of biological fertilizers containing *Bacillus pumilus* strain TUAT-1 and N fertilizer reportedly led to higher tiller numbers of rice at the maximum tillering stage [111]. 

Abiotic stresses, such as salinity and drought, pose major threats to rice growth and yield. Interestingly, *Bacillus amyloliquefaciens* strain NBRI-SN13 isolated from the alkaline soil of Banthara, Lucknow, was found to possess PGPB activities and improve stress tolerance in rice [112,113]. Tiwari et al. reported that *B. amyloliquefaciens* strain NBRI-SN13 positively modulated stress-responsive gene expressions, such as dehydrin (*DHN*) and late embryogenesis abundant (*LEA*), under various abiotic stresses (salt and heat) and phytohormone (abscisic acid) treatments [10]. The results suggested that PGPB play multifaceted roles in crosstalk among stresses and phytohormones in rice especially in osmolyte biosynthesis and subsequently osmotic adjustment. Recently, inoculation of salt-tolerant PGPB, namely, *Bacillus tequilensis* strain UPMRB9 and *Bacillus aryabhattai* strain UPMRE6 on rice plants were shown to have beneficial effects on photosynthesis, transpiration, and stomatal conductance [114]. Shultana et al. demonstrated that the inoculation of *B. tequilensis* strain UPMRB9 on the MR297 rice variety improved total chlorophyll content by 28% and reduced electrolyte leakage by 92% [115]. Increments of relative water content and reduction in the Na/K ratio were also found upon inoculation of *B. tequilensis* strain UPMRB9 and *B. aryabhattai* strain UPMRE6 on rice plants. The results suggested a synergistic effect between PGPB and rice plants on the mechanisms of the plant’s salt tolerance, suggesting the application of PGPB for salinity mitigation practice for coastal rice cultivation. Moreover, the potential application of *Bacillus* to mitigate drought stress in rice has also been demonstrated [116]. Inoculation of rice with drought-tolerant *Bacillus altitudinis* strain FD48 found an increased relative water content, chlorophyll stability index, and membrane stability index in rice. 

In plant disease management, *Bacillus* controls the proliferation of phytopathogens by suppressing plant immunity [117,118]. Suppression of plant immunity by PGPB, referred to as ISR, is one of the important mechanisms to secure the plant against phytopathogens. ISR is defined by the systemic protection of plants by the enhancement of the plant’s defensive capacity against various phytopathogens, which is acquired after appropriate inducing by PGPB [100]. The mechanisms by which PGPB triggered ISR are poorly understood. It is believed that the ISR is triggered by inducing agents (elicitors of ISR) such as antimicrobial metabolites produced by PGPB. Once ISR is triggered, further activation of plant antioxidant enzymes, such as phenylalanine ammonia lyase (PAL), peroxidase (PO), polyphenol oxidase (PPO), chitinase, and β-1,3-glucanase, will take place. This will help the plants to mitigate the reactive oxygen species (ROS) level which is a source of oxidative stress during phytopathogens infection [117,119]. It is worth mentioning that the *Bacillus* can inhibit proliferation of phytopathogens as well as enhance plant immunity directly (by producing antimicrobial metabolites) and indirectly (by producing antioxidant enzymes). As confirmed through genomic analysis, the *Bacillus* genome is composed of antimicrobial metabolite gene clusters (e.g., surfactin and fengycin) and gene-encoding proteins (e.g., PO, PPO, chitinase, and β-1,3-glucanase) that function to suppress plant immunity [117,120]. The plant immune response may be triggered through specific bacterial elicitors produced by *Bacillus*. Further studies should be conducted on the *Bacillus* antimicrobial compounds and antioxidant enzymes beyond in silico genome analysis to understand its contribution in mediated rice plant ISR.

*Bacillus* spp. are known to activate ISR. This has been verified in vitro that rice seeds treated with *Bacillus* spp. showed an elevation of ISR in rice against *X. oryzae* pv. *oryzae* [121]. *Bacillus*-treated seeds exhibited an increased synthesis of defense-related enzymes including PAL, PO, and PPO. On another related study, the induction of systemic resistance against a fungal pathogen, namely, *Rhizoctonia solani* (the causal agent of Sheath Blight) was detected through an increased level of PAL and PO in rice treated with *B. subtilis* [122]. The treatment of *B. subtilis* on rice leaves under greenhouse conditions triggered the accumulation of pathogenesis-related proteins (thaumatin and *β*-1-3-glucanases) that play important roles for the induction of resistance in rice plant.

It is well known that the activity of *Bacillus* spp. as PGPB is linked to their ability to suppress phytopathogens by secretion of antimicrobial metabolites [123]. The secretion of antimicrobial metabolites including surfactins, difficidin, and bacilysin from *Bacillus* spp. trigger the pathways of ISR, which contributes to the suppressive effect of plant immunity [124,125]. Antimicrobial metabolites were determined to act as elicitors of plant immunity and enhance resistance towards further pathogenesis in plants [126]. Sarwar et al. found that purified surfactins from *Bacillus* strains, NH-100 and NH-217, were effective against rice bakanae disease [125]. In 2020, C15surfactin A produced by *Bacillus velezensis* strain HN-2 displayed antibacterial activities against *X. oryzae* pv. *oryzae* and effectively inhibited its infection on rice [119]. It is worth mentioning that the suppression by purified surfactins from *B. amyloliquefaciens* in bean plants was determined to enhance the plant’s ISR against a fungal pathogen, *Botrytis cinerea*, infection [127]. Similarly, in tobacco, surfactins were also found to induce early plant-defense mechanisms [128]. Furthermore, in vitro assays demonstrated the ability of difficidin and bacilysin from *B. velezensis* strain FZB42 (previously *B. amyloliquefaciens* strain FZB42) to suppress rice diseases caused by *Xanthomonas* [129]. The results found that difficidin and bacilysin caused downregulated expression of genes involved in *Xanthomonas* virulence, cell division, protein synthesis, and cell wall synthesis.

### 4.2. Pseudomonas spp.

*Pseudomonas* spp. are Gram-negative, polar-flagellated, and rod-shaped bacteria. This bacterial genus belongs to the phyla Proteobacteria, class Gammaproteobacteria, and family Pseudomonadaceae. Species of *Pseudomonas* and their products have been used in large-scale for biotechnological applications [130]. *Pseudomonas* spp. are ubiquitous in agricultural soils and have many plant growth-promoting traits. Moreover, *Pseudomonas* is a notable bacterial genus because some species are known as clinically important opportunistic human pathogen, plant pathogen, and biocontrol agent. Notable examples include the human pathogen, *Pseudomonas aeruginosa* [131], the plant pathogen, *Pseudomonas syringae* [132], and the non-pathogenic biocontrol agents, *Pseudomonas putida* and *Pseudomonas fluorescens* [133,134]. 

*P. aeruginosa* strains have been intensively studied as an opportunistic human pathogen, especially in an immunocompromised host (a host with a weakened immune system). Interestingly in biocontrol applications, the use of *P. aeruginosa* strains as PGPB has been demonstrated. Strains of *P. aeruginosa* have been reported to be used in seed treatments of rice [135]. All *P. aeruginosa* strains showed plant growth-promoting activity and ISR in rice. Pathogenesis-related peroxidases that are involved in ISR in rice plants were detected in all the *P. aeruginosa* strains and showed antifungal activity against phytopathogenic fungi (*R. solani*, *Pyricularia oryzae*, and *Helminthosporium oryzae*). In 2017, *P. aeruginosa* strain BRp3 (identified using 16S rRNA gene sequencing) isolated from rice rhizosphere was found able to solubilize phosphorus (97 μg/mL) and produced IAA (30 μg/mL) and siderophores (15 mg/L) in vitro [136]. MS analysis revealed the production of siderophores (1-hydroxy-phenazine, pyochellin, and pyocyanin), 4-hydroxy-2-alkylquinolines, rhamnolipids, 2,3,4-trihydroxy-2-alkylquinolines, and 1,2,3,4-tetrahydroxy-2-alkylquinolines in crude extracts of strain BRp3. The results suggested that the secondary metabolites produced by strain BRp3 contribute to its antibacterial activity against *X. oryzae* pv. *oryzae* and its potential to promote the growth and yield of Super Basmati rice.

Strains of *P. syringae* are noted for their diverse and host-specific interactions with a plethora of plant species including rice [51,137]. As a plant pathogen, *P. syringae* colonizes plant tissues by entering plant leaves through the stomata, multiplies in the intercellular space (apoplast), and eventually produces necrotic lesions that are often surrounded by chlorotic halos [138]. However, in 2019, *P. syringae* pv. *syringae* strain 260-02 was found exhibiting non-pathogenic behavior [139]. Strain 260-02 was reported to promote plant growth and exerted biocontrol of *P. syringae* pv. *tomato* strain DC3000, against the *B. cinerea* fungus and the *Cymbidium* ringspot virus. Controversially, the pathogenic status of *P. syringae* has been rising. Considering the genome plasticity of strain 260-02 that could switch to pathogenic behavior through horizontal gene transfer mechanisms, the introduction of this strain into an ecosystem as a biocontrol agent is extremely dangerous.

The implementation of various species of *Psuedomonas* as PGPB on rice plant is well reported [140,141]. For instance, PGPB activities of *P. putida* strain RRF3 have been demonstrated through transcriptomic analysis of rice plant roots [133]. Overall, the results suggested that strain RRF3 immunizes rice plants by re-organizing the root transcriptome to stimulate plant defense responses, and simultaneously protects itself (being a foreign organism) from the primed plants by altering the rhizodeposits. In another related study, the application of a microbial consortium containing *P. putida* (bacterium) and *Chlorella vulgaris* (algae) has proven to ameliorate arsenic toxicity in rice [142]. The positive responses were attributable to a significant decline in arsenic accumulation in root (94 mg kg^−1^ dw) and shoot (51 mg kg^−1^ dw) in a consortium of inoculated seedlings as compared to arsenic exposed plants (156 and 98 mg kg^−1^ dw, respectively). These results suggested that this consortium might alleviate arsenic stress and improve growth of rice seedlings along with a reduction in arsenic levels. Moreover, in vitro characterization of plant growth-promoting activities of *Pseudomonas pseudoalcaligenes* showed that this bacterium has a higher phosphate solubilization and productions of ACC deaminase, siderophores, IAA, and gibberellins [143]. In addition, *P. pseudoalcaligenes* suppressed *Magnaporthe grisea* (the causal agent of Rice Blast Fungus) infection by producing lytic enzymes, namely, chitinase and *β*-1, 3-glucanase. In greenhouse studies, inoculation of *P. pseudoalcaligenes* resulted in the improvement of dry weight, plant height, and root length of rice plants compared to inoculation of *B. pumilus*. 

PGPB strains belonging to fluorescent *Pseudomonas* such as *P. fluorescens* are commonly isolated from rice rhizosphere [134]. For instance, 10 strains of *P. fluorescens* isolated from rice rhizosphere soils in Karnataka, India possesses antibacterial activities against *X. oryzae* pv. *oryzae* [144]. All 10 strains were positive for plant growth-promoting activities including phosphate solubilization and productions of siderophores, HCN, IAA, chitinase, *β*-1, 3-glucanase, cellulase, and salicylic acid. One *P. fluorescens* strain, namely, Pf9, was found to effectively control the causal agent of Bacteria Leaf Blight (BLB) disease. In untreated controls, BLB disease incidence was 80% and this was reduced to 20% in plants raised from strain Pf9 treated seeds. Moreover, biological control of *X. oryzae* pv. *oryzae* by plant-associated *P. fluorescens* producing antimicrobial compound, DAPG, also has been reported [145]. DAPG was found able to inhibit the growth of *X. oryzae* pv. *oryzae* in laboratory assays and reduced BLB disease in net-house (59%) and field (64%) experiments. In 2016, the application of *P. fluorescens* (PGPR_Pf_) in Giza 179 rice cultivar has been tested in the nursery and field [110]. Inoculated rice in the nursery by PGPR_Pf_ exhibited an increase in seed germination, seedling vigor, and yield. In addition, the application of inorganic nitrogen in combination with PGPR_Pf_ (46 kg nitrogen fed^−1^ + PGPR_Pf_ soil application and PGPR_Pf_ foliar spray bacterial application) in the field improved rice yields. 

### 4.3. Enterobacter spp.

*Enterobacter* spp. are Gram-negative, rod-shaped, and non-spore-forming bacteria belonging to the phyla Proteobacteria, class Gammaproteobacteria, and family Enterobacteriaceae. The potential of *Enterobacter* to contribute to the development of sustainable agricultural systems as PGPB has been reviewed previously [146]. However, the mechanisms of PGPB-mediated enhancement of plant growth and yield of many crops are not yet fully understood. It is suggested that *Enterobacter* functions in three different ways: (i) synthesizing particular compounds for the plants, (ii) facilitating the uptake of certain nutrients from the soil, and (iii) lessening or preventing the plants from diseases. These can be characterized by the determination of antagonistic and plant growth-promoting activities [147].

*Enterobacter cloacae* strain B8 [148] and strain B8x [149] have been proven to be effective in improving rice growth and showed antagonistic activity towards *X. oryzae* pv. *oryzae*. In 2020, two bacterial strains (BSB1 and BCB11) isolated from the field showed antagonistic activities towards *B. glumae* were identified belonging to the genus *Enterobacter* [21]. The strains showed high similarity (99%) to *Enterobacter tabaci* as analyzed based on 16S rRNA gene sequences and phylogenetic analyses. Strain BSB1 proved to be the best inorganic phosphorus solubilizer with a solubilization index (SI) of 4.5. Noteworthy, the EtOAc extract of strain BCB11 was found to inhibit the growth of *B. glumae* strains by 85–95%. Metabolomic analysis of EtOAc extract based on GC–MS showed that the main compound present is 3-phenylpropanoic acid (46.7%). This compound showed antibacterial activity with a minimum inhibitory concentration (MIC) of 1000 mg/L against five strains of *B. glumae*. The results suggested that *Enterobacter* is a promising strain as a PGPB and as a source of compound that could inhibit the growth of *B. glumae*.

Up until now, multi-heavy-metal-resistant strains of *Enterobacter* have exhibited positive effects on rice plant growth. In 2018, multi-heavy-metal-resistant-PGPB isolated from metal-contaminated rice rhizosphere were identified as *Enterobacter* sp. strain K2 using phenotypic characterization and MALDI-TOF MS [150]. Strain K2 was found able to resist a group of heavy metals/metalloids (cadmium, lead, arsenic, nickel, and mercury) and possesses plant growth-promoting activities, such as phosphate solubilization, nitrogen fixation, IAA production, and ACC deaminase production. In vitro growth enhancement of a rice cultivar by strain K2 was investigated using cadmium stress and an almost 40% increase in the germination percentage was observed. Other related studies also characterized cadmium-resistance *Enterobacter* that conferred cadmium-tolerance in rice seedlings and are potentially to be applied as PGPB in contaminated fields. Mitra et al. reported that *Enterobacter* sp. strain S2 showed multiple heavy metal resistance on cadmium (3500 µg/mL), lead (2500 µg/mL), and arsenic (1050 µg/mL) [13]. Strain S2 also possesses plant growth-promoting activities based on its ability to solubilize phosphate (73.56 ppm), fix nitrogen (4.4 µg of nitrogen fixed/h/mg protein), produce ACC deaminase (236.11 ng *α*-keto-butyrate/mg protein/h), and IAA (726 µg/mL). Inoculation of strain S2 with rice seedlings significantly enhanced various morphological and biochemical characters of seedling growth compared to un-inoculated seedlings under cadmium stress. 

Moreover, a multi-heavy-metal-resistant-PGPB strain identified as *Enterobacter aerogenes* strain K6 (based on 16S rDNA gene sequence, MALDI-TOF MS, and FAME analyses) was isolated from rice rhizosphere contaminated with a variety of heavy metals/metalloid near an industrial area [151]. Strain K6 exhibited a high degree of resistance to cadmium (4000 μg/mL), lead (3800 μg/mL), and arsenic (1500 μg/mL). Moreover, strain K6 showed several important plant growth-promoting activities, such as phosphate solubilization, nitrogen fixation, IAA production, and ACC deaminase activity under high cadmium stress (up to 3000 μg/mL). Strain K6 was manifested to improve growth of rice seedlings under cadmium stress by lowering oxidative stress (through antioxidants), ethylene stress, and cadmium uptake in seedlings. Furthermore, *Enterobacter* has also been reported to be resistant to chromium (Cr(VI)) and possesses plant growth-promoting activities. For example, *Enterobacter cloacae* strain CTWI-06 was shown to be resistant to 3500 ppm of Cr(VI) [152]. Under optimized conditions, strain CTWI-06 reduced 94% of Cr(VI) within 92 h and reduction was proven by FTIR and XRD analyses. Plant growth-promoting activities such as phosphate solubilization, nitrogen fixation, IAA production, antifungal activities (*R. solani* ITCC 2060 and *Phytium debaryanum* ITCC 5488) were recorded, as well as improved productivity of Mahalakshmi rice in pot culture. The results suggested the potential application of Cr(VI)-reducing strain, CTWI-06, as a bioremediation agent of Cr(VI) in chromium-contaminated soil.

Current explorations of *Enterobacter* spp. are not only focusing on the determination of heavy-metal-resistant-PGPB. Several studies also reported the potential of *Enterobacter* to enhance tolerance of rice plants to salt in soil. For instance, inoculation of *Enterobacter* sp. strain SE-5 resulted in an increment of mature rice plant biomass under salt and cadmium stresses [153]. Strain SE-5 is proposed as an inheritable endophyte due to its ability to be transmitted into rhizosphere, roots, stems, and leaves of mature rice plants as analyzed using green fluorescent protein (*gfp*). In addition, strain SE-5 was found to survive in different soil layers for more than 90 days. Overall, the results suggested that strain SE-5 is potentially proliferating-transmitting in mature rice plants and rhizosphere soil during plant growth. Moreover, salt-tolerant PGPB isolated from rice fields, namely, *Enterobacter* sp. strain P23, were also shown to promote rice seedling growth under salt stress [154]. This effect was correlated with a decrease in plant ethylene production. Reduction in plant ethylene production after inoculation of strain P23 was linked to the bacterial ACC deaminase activity. Strain P23 utilized ACC as a nitrogen source, thus preventing plant ethylene production. In another related study, *Enterobacter* was also found exhibiting ACC deaminase activity [155]. Inoculation with the wild (*Enterobacter* sp. E5) and engineered (*Enterobacter* sp. E5P; overexpressed with ACC deaminase gene) strains promoted the growth of sprouts. The promoting effects were more profound with engineered strain than with wild strain. The engineered strain improved saline resistance of sprouts under salt concentrations from 10 to 25 gL^−1^ and promoted longer roots and shoots than the wild strain. These results suggested that bacterial ACC deaminases play a role in plant-produced ACC degradation and thus inhibit plant ethylene production.

### 4.4. Streptomyces spp.

*Streptomyces* spp. are the most attractive bacterial genus within the scientific community due to their capability to produce various bioactive compounds, which are consequently invaluable in the medical and agricultural fields. *Streptomyces* spp. received a worldwide attention due to their potential as producers of extracellular enzymes [156] and as important sources of secondary metabolites such as antibiotics [157]. *Streptomyces* spp. are complex filamentous Gram-positive bacteria belonging to the phyla Actinobacteria, class Actinomycetes, and family Streptomycetaceae. 

Bacteria from the genus *Streptomyces* are complexly reproduced. Unlike most bacteria that divide by binary fission, *Streptomyces* spp. grow as a mycelium of branching hyphal filaments and reproduce in a mold-like manner by sending up aerial branches that turn into chains of spores [158]. The complexity of *Streptomyces* spp. can also be observed through their large genome size (more than 8 Mbp with a high G + C content), which is often associated with their ability to survive in various environments [159,160]. Interestingly, genome analysis via genome-mining approach has allowed the prediction of biosynthetic gene clusters related to the plant growth-promoting activities of *Streptomyces* strains as a biocontrol agent [161]. 

*Streptomyces* spp. were reported to aid in plant development as PGPB [162,163,164,165]. The characterization of *Streptomyces* spp. as PGPB has provided information on their beneficial traits related to the antagonistic and plant growth-promoting activities [166]. Bacteria belonging to the *Streptomyces* clade isolated from the Colombian Caribbean Sea showed antibacterial activities against *B. glumae*, *B. gladioli*, and *B. plantarii* [167]. Moreover, *Streptomyces* spp. have also been reported to exhibit antagonistic activities against *X. oryzae* [168,169], *X. oryzae* pv. *oryzicola* [170,171], *X. oryzae* pv. *oryzae* [172,173,174], *B. glumae* and *B. gladioli* [175], *P. fuscovaginae* [166], *E. chrysanthemi* [176], and *D. zeae* [177]. 

Genomically, the key pathways relating to plant growth-promoting activity, such as siderophores, IAA, HCN, chitinase, and cellulase, have also been decoded in the genome of *Streptomyces* strains [161]. It is noteworthy that *Streptomyces* spp. are reportedly able to produce antimicrobial metabolites, including blasticidin-S [178], kasugamycin [179], polyoxins [180], and oligomycins [181], that were observed to be active against rice pathogens. The secretion of antimicrobial metabolite from PGPB was suggested to trigger the pathways of ISR in plants which contribute to the suppressive effects of plant immunity [182] and enhance resistance towards further pathogenesis in plants [126]. Other than antimicrobial metabolites, *Streptomyces* spp. also produce a large number of other bioactive metabolites, including VOCs that stimulate plant growth both directly and indirectly [183,184]. Much more focus is still needed to understand the function of antimicrobial metabolites and VOCs from *Streptomyces* spp., particularly with regard to the antagonistic activity against bacterial rice pathogens as well as the ISR of rice plants.

### 4.5. Other Bacterial Genus

Bacterial genera belonging to the phyla Proteobacteria, including *Acidovorax* [185], *Rhizobium* [141,186], *Burkholderia* [187,188], *Serratia* [189], *Azotobacter* [190], *Klebsiella* [188], *Alcaligenes* [191], *Ochrobactrum* [191], *Pseudacidovorax* [192], *Azospirillum* [192,193], and *Herbaspirillum* [192,193], have all been proven to be effective in improving rice growth and yield. Other bacterial genera belonging to the phylum Proteobacteria (*Acinetobacter* and *Pantoea*) and Firmicutes (*Staphylococcus*, *Oceanobacillus*, and *Paenibacillus*) have been reported as PGPB and showed antagonistic effects against *X. oryzae* pv. *oryzae* [194]. Moreover, PGPB with antifungal and antibacterial activities were detected from endo- and rhizospheric bacteria isolated from Basmati rice [195]. Bacterial species belonging to the phylum Proteobacteria (*Stenotrophomonas maltophila* UKA-72 and *Rhizobium radiobacter* UKA-24) and Firmicutes (*B. pumilus* UKA-27) exhibited antimicrobial activities against fungal (*Sclerotium rolfsii*, *F. oxysporum*, and *Rhizoctonia bataticola*) and bacterial (*Xanthomonas compestris* pv. *phaseoli* M5, *X. oryzae*, *Xanthomonas compestris* pv. *phaseoli* CP-1-1, and *Ralstonia solanacerum*) pathogens.

So far, omics technologies have led to the exploration of crop rhizobiome through the metagenomics approach to understanding plant–microbe interactions [196]. Such interactions lead to the selection of plant beneficial microbes, such as PGPB. Interestingly, the population of bacteria in rice rhizosphere was explored metagenomically. In 2020, the population of bacteria in the rhizosphere and phyllosphere of Basmati rice was reported [197]. Bacterial population associated with the rice rhizosphere from three different rice growing areas (Faisalabad, Gujranwala, and Sheikhupura) of Punjab, Pakistan, were compared. Data analyses revealed that Proteobacteria was the dominant phylum at all three areas. In the phyllosphere, Proteobacteria (79.6%) was detected as the dominant phylum followed by Firmicutes (9.8%), Bacteroidetes (8.6%), Chloroflexi (4.3%), and Actinobacteria (0.9%). In other related studies, the 16S rRNA gene amplicon-based metagenomic signatures of the rhizobiome community in rice fields in India have also been evaluated [198]. The results demonstrated that the Proteobacteria (25.69%) is the most abundant bacterial phylum associated with the rice rhizosphere followed by Firmicutes (20.82%), Actinobacteria (16.68%), and Acidobacteria (13.28%). In particular, much more focus is still needed to understand the biological roles of microbes in rice rhizosphere. Therefore, it is necessary to conduct further studies to determine in vivo metabolic activities and the physiological characteristics of rice rhizosphere microbial community. Understanding such attributes will help to shed light on the functionality as well as biological roles of microbes in rice rhizosphere not only for improved plant health as biofertilizers but as biological agents to combat phytopathogens.

## 5. Bioformulations of PGPB in Rice: Applications, Challenges, and Future Prospects

### 5.1. Applications on Bioformulations

Bioformulations are defined as any biologically active substances derived from microbial biomass or products containing microbes and microbial metabolites that are used in plant growth promotion, nutrient acquisition, and disease control in an eco-friendly manner [199]. The application of PGPB in bioformulation development has been reviewed previously [164,200,201]. In the development of bioformulations, three potent components are needed, which include: (i) active ingredient, (ii) carrier material, and (ii) additive [202]. The active ingredient is typically a viable organism (live microbes or spores that are able to survive during storage), while carrier material is an inert substance that supports the active ingredient (cells). The carrier material assures that cells are able to easily proliferate in or around the plant and to provide better chances of enhancing biocontrol and plant growth-promoting activities. Carrier materials such as talc, charcoal, wheat bran, rice husk, saw dust, fuller’s earth, and sugarcane bagasse were found to increase the shelf life of the active ingredient [203,204]. Additives such as gum arabic, trehalose, glycerol, alginate, and carboxymethyl cellulose were reported protect the cells and provide a longer shelf life along with providing tolerance from harsh environmental conditions, while improving physical, chemical, and nutritional properties of bioformulations [205].

Many reports suggested that bioformulations are easy to deliver, able to enhance plant growth and stress resistance, and able to increase plant biomass and yield. Additionally, any bioformulation with an increased shelf life with simultaneous actions of biocontrol and biofertilizer activities under field conditions could open the way for technological exploitation and marketing [206,207]. With rice, PGPB bioformulations have been applied to combat bacterial as well as fungal rice pathogens [17,208,209,210,211]. In 2017, *P. aeruginosa* strain BRp3 has been suggested as a bio-inoculant for Super Basmati rice and showed antagonistic activities against *X. oryzae* pv. *oryzae* [136]. The inoculation of strain BRp3 supplemented with 80% of the recommended doses of N and P (140–80 kg NP acre^−1^) significantly enhanced grain and straw yield with an increase of 51% and 55%, respectively. Colonization studies under field conditions using viable count detected the colonization of strain BRp3 on rice roots and shoots up to 60 days, suggesting its application as rhizobacteria-inoculants. 

In another related study, the efficacy of PGPB, *P. fluorescens*, as a bioformulation in rice has been reported. Seed treatment with bioformulation of *P. fluorescens* strain RRb 11 increased the plant growth-promoting parameters of Pusa Basmati 1 rice [15]. The talc-based bioformulation not only reduces disease intensity (caused by *X. oryzae* pv. *oryzae*) but enhances germination, increases height, dry matter, and yield. In addition, Jambhulkar and Sharma showed antagonistic potential of *P. fluorescens* strain RRb-11 against *X. oryzae* pv. *oryzae* [16]. The results stated that the maximum shelf life of strain RRb-11 was recorded up to 150 days after storage in talc-based bioformulations. In field studies, the bioformulation was applied as seed treatment, seedling root dip, and soil application in combination. Overall, the reduction in BLB disease against control was recorded by 92.3% in the year 2009 and 88.5% in the year 2010. The treatment also produced a maximum yield (61%) greater than control. The bioformulation of PGPB, *P. fluorescens* strain SP007s, has also been applied in Thailand [212]. The application of strain SP007s as a bioformulation termed as ISR-P/K via seed treatment, broadcasting, and foliar spray significantly achieved the best results in yield improvement at 52.1% and a reduction in bacterial (*X. oryzae* pv. *oryzae*) and fungal (*Pyricularia grisea*, *H. oryzae*, *Cercospora oryzae*, *R. solani*, *Curvularia lunata*, *Fusarium semitectum*, *Alternaria padwickii*, and *Sarocladium oryzae*) pathogens.

The use of PGPB in the form of bioformulation is an eco-friendly and inexpensive alternative to chemical fertilizers and pesticides [213,214]. It is worth mentioning that several microbe-based products or microbial consortia to improve rice yields are commercially available to farmers worldwide [215,216,217,218]. These include Bio-N (Nutri-Tech Solutions, Yandina, Queensland, Australia) containing *Azospirillum* spp.; BioGroe^®^ (Nguyen Thanh Hien in Hanoi University, Hanoi, Vietnam) containing *P. fluorescens*, *B. subtilis*, *B. amyloliquefaciens*, and *Candida tropicalis*; Dimargon^®^ (Bernardo Dibut Alvarez in INIFAT, Havana, Cuba) containing *Azotobacter chroococcum*; EMAS (Didiek Hadjar Goenadi in IRIBB, Bogor, Indonesia) containing *Azospirillum lipoferum*, *Acinetobacter beijerinckii*, *Aeromonas punctate*, and *Aspergillus niger*; BioPower (NIBGE, Faisalabad, Pakistan) containing multiple strains of nitrogen-fixing bacteria.

### 5.2. Challenges and Future Prospects

To enhance yield and protect crops against pests and pathogens, the use of chemical fertilizers and pesticides have been crucial in ensuring food security to feed the ever-increasing human population [219]. Repeated use of chemical fertilizers and pesticides that are rich in nitrogen, phosphorous, and potassium lead to soil, air, and groundwater pollution [220]. Interestingly, the use of PGPB is a potent and upcoming method to ensure sustainable agriculture without depleting natural resources [221,222]. The introduction of successful PGPB to soil ecosystems improves soil’s physical (e.g., reducing sodicity and bulk density, improving water infiltration rate, and increasing porosity and aeration) and chemical (e.g., reducing acidity) properties [223]. However, before PGPB can be applied to the environment, there are several safety standards and qualities that should be fulfilled [224,225]. A potential PGPB should: be identified as a taxonomical unit univocally, be effective against target phytopathogens, not show clinical or animal toxicity, and not persist in the agro-environment. Moreover, PGPB should not transfer its genetic material to other closely related microbes to avoid the risk of antibiotic resistance development [226]. With these, the antagonistic potential and behavioral features of a potential PGPB must be thoroughly characterized to permit its registration as a biocontrol agent and approval for use in plant protection. 

The introduction of the research results for industrial exploitation is not necessarily easy. Several PGPB bioformulations published worldwide demonstrated outstanding biocontrol activities in vitro [11,227]. Often a lack of field results failed to support the applicability of PGPB in commercial fields/greenhouses studies. This limitation hinders the commercial development of successful PGPB bioformulations. Furthermore, the biocontrol activity of PGPB is also hindered due to the fact that the production of antimicrobial metabolites by PGPB is strictly dependent on the PGPB’s culturing substrate as well as abiotic and biotic stresses around them [228,229]. Additionally, the potential of PGPB to control newly emerging bacterial rice pathogens remains obscure. Further extensive studies on plant–microbe interaction mechanisms (beyond the rhizosphere), especially on biocontrol and plant growth-promoting actions, are still not sufficiently explained. Advances of the omics technologies through next-generation sequencing (NGS) and molecular biology studies such as metagenomics, metabolomics, proteomics, and culturomics will be necessary to understand the fate of PGPB in mediated plant–microbe interactions [230,231]. Shedding light on the symbiotic interaction of PGPB with rice might lead to the development of highly effective and efficient bioformulation across different soil types and environmental conditions. Overall, much more focus is still needed to fulfill the industrial demands for the production of effective bioformulations with one or more active ingredients, using different carrier materials and additives, and with various methods/treatments of field inoculations. 

## 6. Conclusions

A plethora of studies have addressed the ability of PGPB to promote plant growth on rice plants and its biocontrol activity against bacterial rice pathogens. As this review has made clear, bacterial rice pathogens can be caused by a variety of bacterial genus, including *Xanthomonas*, *Burkholderia*, and *Pantoea*. Interestingly, PGPB could be a possible alternative in controlling bacterial rice pathogens, which involves the application of eco-friendly microbes that control pathogens and improve plant growth. With the awareness on the usage of PGPB in the agricultural sector, their isolation and characterization are highly demanded, especially for use in rice plant protection. Varieties of bacterial genera, including *Bacillus*, *Pseudomonas*, *Enterobacter*, and *Streptomyces*, are promising as rice plant co-inoculants. Their application in the form of bioformulations could potentially improve the sustainable production of rice. The use of a bacterium or a consortium of PGPB in correct bioformulations provides a remarkable solution for a more sustainable agricultural future. In conclusion, the studies mentioned in this review support the progress in characterizing PGPB and designing bioformulations for use in rice plant protection to manage bacterial rice pathogens. In addition, this review highlighted the importance of continuing research on the applicability of PGPB, which, up to now, are scarcely used as rice plant co-inoculants to enhance rice production and ensure food security.

## Figures and Tables

**Figure 1 microorganisms-09-00682-f001:**
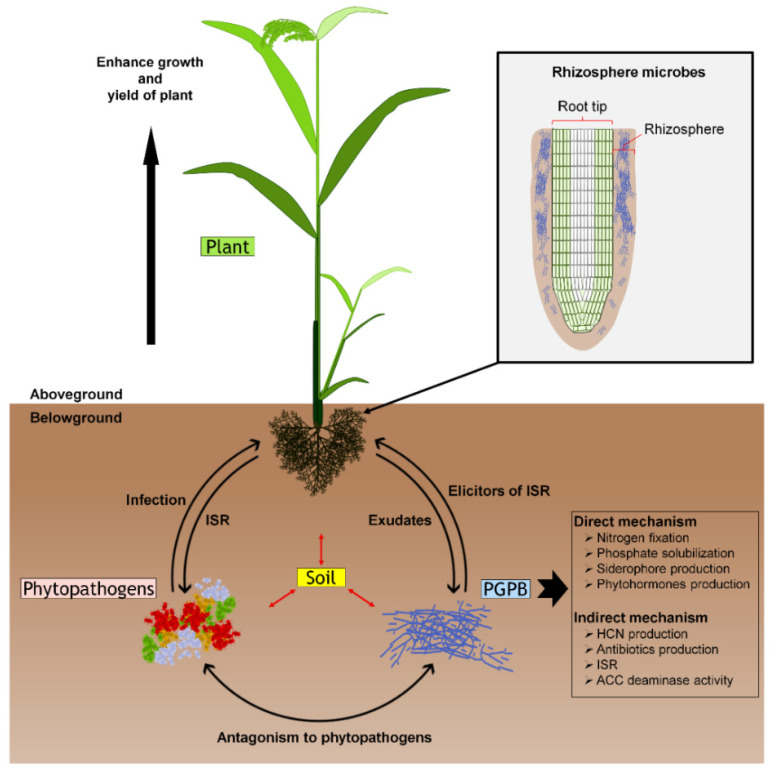
Biological control interactions exerted by the plant growth-promoting bacteria (PGPB). This illustration depicts the interactions between PGPB, phytopathogens, and plants. PGPB promote plant growth either by direct and or indirect mechanisms. PGPB colonize plant’s rhizosphere and produce antimicrobial metabolites. In the plant’s rhizosphere, antibiosis and nutrient competition interaction suppresses the growth of phytopathogens. Elicitors of induced systemic resistance (ISR) production by PGPB and in the simultaneous presence of phytopathogens enhanced the plant ISR. Thus, this mediated defense response of plants towards phytopathogens and consequently enhanced plant growth and health.

**Table 2 microorganisms-09-00682-t002:** In vitro characterizations of promising PGPB.

Mechanisms		Media	Descriptions	References
**Direct**				
Nitrogen fixation		Nitrogen-free (NF) agar	Nitrogen fixation was observed qualitatively by the blue coloration around the colonies	[80]
		Malate (NFM) semisolid medium	Acetylene production was quantified on a gas chromatograph equipped with a Porapak Q column and a H_2_-flame ionization detector (FID)	[81,82]
Phosphate solubilization		Pikovskaya’s agar	Phosphate solubilization was determined qualitatively by the formation of halo zones around the colonies	[83]
Siderophore production		Chrome azurol S (CAS) agar	Siderophore production was observed qualitatively by the yellow halo coloration around the colonies	[84]
Phytohormones production	IAA	Nutrient broth medium supplemented with _L_- tryptophan	IAA production was determined using colorimetric methods and quantified on HPLC using ethyl acetate oxidation method	[85,86,87]
	Cytokinins	Burk’s medium	Cytokinin production was determined using colorimetric methods	[88,89]
	Gibberellins	Nutrient broth medium	Gibberellin production was determined using colorimetric methods	[90]
**Indirect**				
ACC deaminase production		Dworkin and Foster’s (DF) salts medium	Colonies growing on the DF agar were taken as ACC deaminase producers and ACC deaminase activity was determined using colorimetric method	[91,92]
HCN production		Nutrient broth supplemented with 4.4 g/L of glycine	HCN production was observed qualitatively by the changes in the filter paper color from yellow to orange-brown	[93,94]
Antibiotics production		Mueller Hinton (MH) medium	Screening of antimicrobial activity was observed using diffusion methods	[95,96]

## Data Availability

Not applicable.

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
