# Peer review of "Plant Growth-Promoting Bacteria as an Emerging Tool to Manage Bacterial Rice Pathogens"

_microorganisms, 2021, doi:10.3390/microorganisms9040682_

Round 1

Reviewer 1 Report

The review “A review on research progress and implementation of plant growth-promoting bacteria as a tool to manage bacterial rice pathogens” highlights the main aspects about the known mechanisms to enhance rice plant growth by PGPB. The topic is interesting to define the PGPB as bioformulation for improving rice productivity.

The work is well written but minor revisions are needed:

  • Title: I think that title is not captivating. A possible title is “ Plant growth-promoting bacteria as an emerging tool to manage bacterial rice pathogens”
  • 234-235: Explain the concept “suppressing plant immunity”. What are the proteins that suppress the plant’s immunity? I think that it is interesting to deal the concept “antimicrobial metabolites enhance resistance to other pathogens”.
  • Moreover you could treat about the secondary metabolites and the antioxidant defense activity in rice induced by PGPB, with reference to gene expression such as PAL, and other genes coding key enzymes for the lignin and phytoalexins, because their important role in abiotic stress response.

Reviewer 2 Report

The article begins by presenting the importance of rice crops and the main phytopathogens  of bacterial origin (Xanthomonas sp., Burkholderia sp., Pantoea sp.) that cause the production losses. An interesting presentation is made for the so-called plant growth promoting bacteria (PGPB) that could be used as bioproducts with biocontrol and biostimulation activity in rice culture. Some qualitative methods are briefly presented, which can highlight the ability of microorganisms to fix nitrogen, to solubilize phosphorus, to produce siderophores and / or phytohormones, with the appropriate bibliographic links for the interested parties in the practical deepening of this subject.
Bacterial strains like Bacillus sp., Pseudomonas sp., Enterobacter sp.and Streptomyces sp. appear to be promising in terms of their use in obtaining bioproducts with biocontrol and biostimulation activity. The development and application of such bioproducts in rice cultivation can improve production and provide a solution for sustainable agriculture, with a positive impact on human health and protection of environmental factors by reducing the use of pesticides, growth stimulants and fertilizers. of synthesis
The paper emphasizes the importance of continuing research regarding the developping and application of bioproducts containing PGPB in rice culture (a method that currently is little used) in order to improve production and ensure food security.
The manuscript encourages the reader to read it until the end and determines him to keep it, because it offers important information, which can be accessed in detail through the extensive bibliography, presented at the end. Personally, I find this article interesting, and I recommend it for publication for three reasons
1) Provides information and explains the mechanism of action of some microorganisms that could be used in rice cultivation, as biocontrol and biostimulation agents;
2) The manuscript offers an approach model for obtaining biopreparations that can be applied to other important agricultural crops in Europe, America, Australia, etc. such as wheat, potato, corn, etc .;
3) Microbial biopreparations containing microorganisms or the consortium of microorganisms that produce growth biostimulators and microbial antagonists can replace pesticides and chemical synthesis products used in agriculture. Replacing them with microbial biopreparations can have a significant positive impact on human health and environmental protection.
